# Modulated Electro-Hyperthermia Accelerates Tumor Delivery and Improves Anticancer Activity of Doxorubicin Encapsulated in Lyso-Thermosensitive Liposomes in 4T1-Tumor-Bearing Mice

**DOI:** 10.3390/ijms25063101

**Published:** 2024-03-07

**Authors:** Kenan Aloss, Syeda Mahak Zahra Bokhari, Pedro Henrique Leroy Viana, Nino Giunashvili, Csaba András Schvarcz, Gábor Szénási, Dániel Bócsi, Zoltán Koós, Gert Storm, Zsuzsanna Miklós, Zoltán Benyó, Péter Hamar

**Affiliations:** 1Institute of Translational Medicine, Semmelweis University, 1094 Budapest, Hungary; kenan.aloss@phd.semmelweis.hu (K.A.); syeda.bokhari@phd.semmelweis.hu (S.M.Z.B.); pedro.leroy@phd.semmelweis.hu (P.H.L.V.); nino.giunashvili@phd.semmelweis.hu (N.G.); schvarcz.csaba.andras@semmelweis.hu (C.A.S.); szenasi.gabor@semmelweis.hu (G.S.); bocsi.daniel@semmelweis.hu (D.B.); koos.zoltan@semmelweis.hu (Z.K.); miklos.zsuzsanna@med.semmelweis-univ.hu (Z.M.); benyo.zoltan@semmelweis.hu (Z.B.); 2HUN-REN-SU Cerebrovascular and Neurocognitive Diseases Research Group, 1094 Budapest, Hungary; 3Department of Pharmaceutics, Utrecht Institute for Pharmaceutical Sciences, Utrecht University, 3508 Utrecht, The Netherlands; g.storm@uu.nl; 4Department of Surgery, Yong Loo Lin School of Medicine, National University of Singapore, Singapore 117597, Singapore; 5National Korányi Institute for Pulmonology, 1122 Budapest, Hungary

**Keywords:** modulated electro-hyperthermia, doxorubicin, lyso-thermosensitive liposome, PEGylated liposome

## Abstract

Modulated electro-hyperthermia (mEHT) is an adjuvant cancer therapy that enables tumor-selective heating (+2.5 °C). In this study, we investigated whether mEHT accelerates the tumor-specific delivery of doxorubicin (DOX) from lyso-thermosensitive liposomal doxorubicin (LTLD) and improves its anticancer efficacy in mice bearing a triple-negative breast cancer cell line (4T1). The 4T1 cells were orthotopically injected into Balb/C mice, and mEHT was performed on days 9, 12, and 15 after the implantation. DOX, LTLD, or PEGylated liposomal DOX (PLD) were administered for comparison. The tumor size and DOX accumulation in the tumor were measured. The cleaved caspase-3 (cC3) and cell proliferation were evaluated by cC3 or Ki67 immunohistochemistry and Western blot. The LTLD+mEHT combination was more effective at inhibiting tumor growth than the free DOX and PLD, demonstrated by reductions in both the tumor volume and tumor weight. LTLD+mEHT resulted in the highest DOX accumulation in the tumor one hour after treatment. Tumor cell damage was associated with cC3 in the damaged area, and with a reduction in Ki67 in the living area. These changes were significantly the strongest in the LTLD+mEHT-treated tumors. The body weight loss was similar in all mice treated with any DOX formulation, suggesting no difference in toxicity. In conclusion, LTLD combined with mEHT represents a novel approach for DOX delivery into cancer tissue.

## 1. Introduction

Doxorubicin (DOX) is one of the most widely used anticancer agents [1]. However, DOX use is limited due to its systemic side effects, such as cardiotoxicity [2] and myelosuppression [3]. Two liposomal DOX formulations have been approved by the Food and Drug Administration (FDA): PEGylated (Doxil^®^, Caelyx^®^, Lipodox^®^) and non-PEGylated (Myocet^®^) liposomal DOX [4]. Although the clinically approved formulations have reduced cardiotoxicity, their efficacy is not superior to free DOX [5,6]. The unchanged efficacy has been explained by the poor DOX release from the liposomes [7], and by the complete reliance on the heterogeneous enhanced permeability and retention (EPR) effect [8].

Doxorubicin exerts its anticancer effects by intercalating into DNA and generating free radicals [9]. Ultimately, DOX results in cancer cell death [10]. DOX-induced apoptosis is mainly a caspase-dependent apoptosis, which is characterized by the elevation of cleaved caspase-3 (cC3), the main effector of apoptosis [11]. Furthermore, DOX inhibits cancer cell proliferation, demonstrated by reduced Ki67 expression, a commonly used proliferation marker [12].

Lyso-thermosensitive liposomal DOX (LTLD) (Thermodox^®^, Celsion Corporation, Lawrenceville, NJ, USA) presents a promising approach to overcome the limitations of the approved liposomal formulations due to an ultra-fast release of DOX at temperatures > 39.5 °C (≈80% within 20 s) [13]. Furthermore, LTLD releases DOX into the bloodstream in the heated tumor, which is followed by DOX diffusion into the tumor interstitium, thereby bypassing the dependence on the EPR effect [14]. LTLD is the first thermo-sensitive liposome (TSL) to reach the clinical trial stage. However, following successful phase I and II clinical trials [15], two phase III clinical trials: “HEAT” (Heat-Activated Target Therapy investigating LTLD addition to local heating with radiofrequency ablation (RFA) of hepatocellular carcinoma (HCC)) [16] and OPTIMA (optimized based on the results of the HEAT study) [17], failed to reach the primary and secondary endpoints in HCC patients. Factors contributing to the outcome of the HEAT study have been described previously [18]. In addition, RFA results in heat diffusion from the ablative zone (T > 50 °C) into the tumor margins (T: 39–40 °C), where DOX is released from LTLD. Between these two regions, an intermediate zone with vascular shunts can be formed, leading to the obstruction of DOX delivery [19]. Therefore, the need for developing new HT modalities to activate LTLD is justified.

Modulated electro-hyperthermia (mEHT) is an advanced option in the hyperthermia field that applies a 13.56-radiofrequency electromagnetic current generated by a capacitive coupling setup between two electrodes [20]. mEHT is approved for cancer therapy in several countries, exhibiting promising results in tumor responses with no serious side effects in different cancer types [21,22,23,24,25]. One great advantage of mEHT is the non-invasiveness [26], in contrast to RFA, which requires the insertion of a needle into the tumor tissue [27]. The tumor-specific damage of mEHT is based on the difference in the bioelectrical properties between the tumor and healthy tissues [28]. This bioelectrical difference results from the higher aerobic glycolysis of cancer cells that causes higher ion and lactate levels and thereby elevates the electric conductivity of the tumor [29,30]. These factors result in the selective absorption of the energy of an electromagnetic field by the tumor tissue [28]. Moreover, the specific absorption rate (SAR), which refers to the rate at which electromagnetic energy is absorbed by body tissues, can be controlled and modulated during the application of mEHT. This control allows for the precise targeting of the tumor area while minimizing the effects on the surrounding healthy tissues, and it thereby contributes to the tumor selectivity of the treatment [31]. Previously, our group demonstrated that mEHT enables a +2.5 °C selective heating of the tumor (ΔT_(tumor-skin)_ ≥ 2.5 °C) [28]. This tumor-selective heating might introduce mEHT as a more efficient induction of DOX release from LTLD within the tumor. Therefore, in the present study, we aimed to investigate the effects of mEHT as a hyperthermia source on the DOX release from LTLD in the tumor and its anticancer activity.

## 2. Results

### 2.1. mEHT Enhanced Tumor Growth Inhibition of LTLD-Encapsulated DOX

Ultrasound and caliper measurements demonstrated steady tumor growth over time in the sham+vehicle group (Figure 1A,B), resulting in the heaviest tumors upon study termination (Figure 1C). Compared to the sham+vehicle, most treatments decreased the tumor growth; however, the mEHT+LTLD-treated mice carried the smallest tumors at the end of the observation period (Figure 1A,B).

Upon tumor removal, tumors were the largest in the sham+vehicle and sham+DOX groups. The sham+PLD, mEHT+vehicle, and mEHT+DOX tumors were smaller, and the mEHT+PLD and mEHT+LTLD tumors were the smallest, as shown in the photos (Figure 1D) and the tumor weight curve (Figure 1C). The mEHT+LTLD tumors were smaller than the mEHT+PLD tumors (Figure 1D); however, this difference was not statistically significant (Figure 1C). mEHT+LTLD was the only treatment that significantly reduced the tumor weight compared to mEHT+DOX (Figure 1C).

A separate experiment was conducted to compare the tumor growth inhibition of PLD and LTLD for 5 days after the termination of the treatments. mEHT+LTLD was significantly more effective at inhibiting tumor growth than mEHT+PLD (Figure 2A–C). Based on the ultrasound recording, this difference was already visible after the second treatment (Figure 2A). Tumors treated with mEHT+LTLD or mEHT+PLD stopped growing already after the second treatment and did not regrow during the following 5-day observation period after the last treatment (Figure 2A). Upon study termination, the mEHT+LTLD-treated tumors were significantly smaller than the mEHT+PLD-treated tumors (Figure 2B,C).

### 2.2. mEHT Enhanced Early Tumor Accumulation of DOX from LTLD-Encapsulated DOX

The in vivo imaging system (IVIS) did not detect any DOX autofluorescence in tumors treated with sham+DOX, sham+PLD, mEHT+DOX, or mEHT+PLD at 1 h after the treatment. However, a strong DOX autofluorescence signal was observed in the mEHT+LTLD-treated tumors, accumulating the most DOX in the tumor 1 h after treatment. After 24 h, a 30% reduction was observed in the DOX autofluorescence signal from tumors treated with LTLD+mEHT. At 24 h, PLD-treated and mEHT+PLD-treated tumors also demonstrated DOX autofluorescence, which was similar to the LTLD-treated tumors (not significant) (Figure 3A,B).

### 2.3. mEHT+LTLD Enhanced Tumor Tissue Destruction

The TDR, estimated as the percentage of damaged area to tissue surface area in hematoxylin and eosin (H&E) sections, revealed low TDRs in the sham+vehicle- and DOX-treated tumors. mEHT significantly increased the TDRs compared to the sham+vehicle, sham+DOX, and sham+PLD groups. The TDRs in the PLD-treated tumors were significantly higher than those in the sham+DOX group. Furthermore, mEHT+PLD increased the TDRs compared to PLD alone. The TDRs in the mEHT+LTLD group were significantly higher than those in all the other groups except mEHT+PLD. Although there was a slight difference, mEHT+LTLD did not result in significantly different TDRs as compared to mEHT+PLD (Figure 4A,B).

### 2.4. mEHT+LTLD Augmented Caspase-Dependent Apoptosis

On the H&E-stained sections (Figure 4A), the damaged areas of the tumors were visible as pale areas, and the consecutive sections were stained brown by immunohistochemistry (IHC) for cC3. The degree of cC3 staining was significantly correlated with the TDR (R^2^ = 0.73, *p* < 0.0001) (Figure 5C), suggesting that apoptosis was involved in the cell death. In the sham+vehicle-treated tumors, the cC3 expression was low. Monotherapy with DOX or PLD did not increase the cC3-stained area over the sham+vehicle. In contrast, mEHT induced a significant increase in cC3 staining compared to the sham or DOX treatments alone. However, combining mEHT with DOX or PLD did not change the degree of cC3 staining. The most intensive cC3 staining was observed in the mEHT+LTLD-treated tumors (Figure 5A,B). Consistent with the IHC results, western blotting of cC3 demonstrated that the cC3 expression in the mEHT+LTLD-treated tumors was significantly higher than that in the tumors in all the other groups (Figure 5D,E).

### 2.5. mEHT+LTLD Alleviated Tumor Cell Proliferation

The Ki67-proliferation-marker-positive nuclei were abundant in the IHC-stained sections of the sham+vehicle-treated tumors, indicating the strong proliferation of tumor cells. Neither DOX nor PLD influenced the proliferation. However, mEHT significantly reduced the number of Ki67^+^ nuclei compared to the sham+vehicle-treated tumors. The number of Ki67^+^ nuclei was significantly less in the mEHT+DOX-treated tumors than in the mEHT- and DOX-treated tumors. The mEHT+PLD-treated tumors had significantly fewer Ki67^+^ nuclei than the PLD- and mEHT-treated tumors. The mEHT+LTLD group had the lowest number of Ki67^+^ nuclei (Figure 6A,B). Similarly, western blotting of Ki67 exhibited that the mEHT+LTLD-treated tumors had the lowest Ki67 expression (Figure 6C,D).

### 2.6. Mice Treated with DOX Lost Body Weight

Animals not receiving DOX in any form (sham+vehicle- and mEHT+vehicle-treated mice) had steady body weights during the study. In contrast, a significant decrease in body weight was observed in all mice treated with any DOX formulation (DOX, PLD, or LTLD) (Figure 7). These groups exhibited parallel body weight loss, and there was no significant difference in the kinetics of the body weight loss between them (Figure 7).

## 3. Discussion

In this study, we investigated the tumor penetration and anticancer effects of doxorubicin encapsulated in lyso-thermosensitive liposomes (LTLD) in mice subjected to modulated electro-hyperthermia (mEHT). The effects of several other doxorubicin formulations were tested for comparison. Doxorubicin release from LTLD has previously been tested using different conventional hyperthermia (cHT) modalities, such as the immergence of mice in a water bath [32], or RFA in pigs [33] and humans [16]. However, the anticancer effects of cHT are based only on the thermal effect, which is only 1 °C ΔT between the tumor and its surrounding tissues [34]. mEHT combines both thermal and non-thermal effects, and a ΔT of 2.5 °C can be achieved [28]. Thus, the anticancer effect of mEHT is superior to those of cHT modalities [35,36], and the greater ΔT may provide a better local release of DOX from LTLD within the tumor. As a result of the non-thermal feature, as well as the higher ΔT, the antitumor effect of mEHT is more specific and can be achieved by reducing heat-related toxicities such as burns, which are often observed after cHT treatment [37]. In a review [38], the hypothesis is proposed that mEHT can be combined with thermosensitive liposomes (TSLs); however, no original research has previously been performed to investigate this hypothesis.

The present study confirmed that the mEHT+LTLD treatment has the strongest tumor growth inhibitory effect. Moreover, mEHT+LTLD had already reduced tumor growth after two treatments compared to mEHT+PLD (Figure 2A). Similar to our study, Needham et al. also found that LTLD activated with water-bath HT reduced tumor growth more than PLD in a human tumor xenograft model [32]. Similarly, Dromi et al. reported the superior tumor growth inhibition with LTLD to that with free DOX and PLD using pulsed high-intensity focused ultrasound (pulsed-HIFU) for the heat activation [39]. In pulsed-HIFU, the tumor is exposed to 2 min pulses of HIFU, leading to a minimal temperature rise (39–44 °C) and non-thermal, mechanical effects demonstrated by local radiation force-induced tissue displacements and consequent shear forces that alter the tissue permeability [40]. However, Wang et al. demonstrated that pulsed-HIFU increased the tumor temperature to 44 °C, but the temperature of the surrounding muscles reached 42 °C [41], which might lead to a significant accumulation of DOX from LTLD in these surrounding tissues.

In our study, the greater inhibition of tumor growth in the mEHT+LTLD-treated mice than in the mEHT+PLD-treated mice is attributed to the acceleration of the DOX local delivery. The highest DOX accumulation in the tumors was observed 1 h after mEHT+LTLD (Figure 3A,B), as expected based on the pharmacokinetic profile of LTLD [14]. In contrast to PLD, LTLD is not dependent on the EPR effect because it releases 80% of DOX into the bloodstream in the heated tumor, which is followed by the diffusion of DOX into the tumor interstitium [14,42]. In contrast, PLD extravasation is mainly based on the EPR effect, which is a relatively slow process [43]. This explains why mEHT induces the early accumulation of DOX in the tumor if delivered by LTLD but not if delivered by PLD. Furthermore, this complete dependence of PLD delivery on the EPR effect explains the improvement in the DOX accumulation in the tumors 24 h after the PLD injection [43]. The reduction in the tumor DOX concentration 24 h after LTLD treatment (Figure 3A,B) was expected due to the wash-out of free DOX [44]. These results are consistent with observations from previous studies on LTLD in mice [45,46].

We found that mEHT did not enhance the DOX accumulation from PLD (Figure 3A,B) in the tumor, as shown in previous studies using a similar degree of hyperthermia (42 °C) [39,47]. However, Tsang et al. reported that mEHT applied 4 h after liposome injection could enhance the cellular uptake of DOX from PLD, whereas we applied mEHT immediately after PLD or LTLD injection [48]. Thus, the timing of the HT can affect the cellular DOX uptake from liposomes.

Our histological examinations demonstrated that treatment with mEHT+LTLD or mEHT+PLD caused the greatest tumor damage (Figure 4A,B), which can be explained by the synergism between mEHT and DOX in inducing apoptosis. Our group previously revealed that mEHT induces caspase-dependent apoptosis [20]. In the current study, low cC3 was detected in the DOX-treated tumors; however, cC3 increased dramatically when different DOX formulations were combined with mEHT, reaching the highest expression after mEHT+LTLD (Figure 5A,B,D,E). Similarly, Maswadeh et al. revealed that TSL+HT was more effective at inducing apoptosis in murine fibrosarcoma tissues than non-TSL+HT [49]. Additionally, an in vitro report revealed the superior effects of PLD over free DOX on apoptosis induction in oral squamous cell carcinoma CAL-27 cells, although the effect of TSL was not measured [50]. In the current study, we did not investigate other mechanisms for cell death, such as autophagy or necrosis. In our previous studies, mEHT did not affect the gene expressions of molecules involved in autophagy, such as Beclin1, autophagy-related protein 3 (ATG 3), ATG 5, and sequestosome 1 (SQSTM1) [20]. Moreover, in hyperthermia treatment, necrosis is mainly activated at temperatures > 45 °C [51,52].

DOX inhibits cancer cell proliferation [53], although free DOX or PLD did not reduce proliferation in the current study. However, a strong proliferation inhibition was observed when DOX was combined with mEHT, especially for the liposomal forms (i.e., PLD and LTLD). Furthermore, LTLD caused stronger proliferation inhibition than PLD (Figure 6A–D). In a previous study, PLD combined with HT inhibited proliferation more than HT or PLD alone in a 4T1 mouse model. However, the effect of TSL was not reported in the study [54].

The present study demonstrated a similar decrease in body weight in mice treated with DOX, PLD, or LTLD (Figure 7). The LTLD membrane contains 10% lysolipids, which can interact with plasma proteins and dissociate from the membrane, leading to leakage from the LTLD at body temperature and systemic toxicity [55,56]. In contrast, the interstitial release of DOX from PLD reduces systemic toxicity [43]. However, in the current study, PLD resulted in significant body weight loss. Besse et al. demonstrated that mice treated with LTLD at 5 and 10 mg/kg had stronger body weight loss than DOX- and PLD-treated mice. In the study, the maximum body weight loss after LTLD treatment was 7% [57]. The reason for the stronger toxicity of LTLD in our study was the repeated dosing (three doses), unlike most previous preclinical studies, in which LTLD was administered in a single dose [32,39,45,57,58,59]. The repeated dosing was selected because chemotherapeutic drugs and mEHT are usually used in cycles in the clinic. In addition, a single dosing of LTLD has been suggested as one of the reasons for the failure of clinical trials [18]. Therefore, LTLD is being administered in cycles in two ongoing clinical trials (NCT02536183, NCT03749850).

The main limitation of the present study is that mEHT+PLD caused strong anticancer effects. Thus, the possibility of improving the anticancer effects by using LTLD was narrow. However, the use of thermosensitive liposomes and the deliberation of their cargo with mEHT offers a new, clinically relevant, alternative possibility, where the tumor-specific delivery of the chemotherapeutic drug is not dependent on the EPR. The EPR effect is notably heterogeneous within the tumor and between tumor types [8]. Furthermore, some tumor types, such as pancreatic cancer, prostate cancer, and metastatic liver cancer, have poor EPR due to the thick fibrous stroma and hypovascularity, which hamper the penetration of nanoparticles [60,61].

## 4. Material and Methods

### 4.1. Free DOX and DOX Liposomes

LTLD (Thermodox, frozen) was obtained from Celsion Cooperation (Lawrenceville, NJ, USA). Each vial contained 15 mL of LTLD (2 mg/mL doxorubicin hydrochloride). After thawing, LTLD was divided into 3 mL aliquots to avoid freeze–thaw cycles and frozen at −80 °C. PLD (Caelyx, 2 mg/mL) was provided by Baxter Holdings (Utrecht, The Netherlands). Both LTLD and PLD were diluted with 0.9% NaCl before administration. Doxorubicin hydrochloride was purchased from MedChem Express (Monmouth Junction, NJ, USA, Cat. No. HY-15142) and dissolved in 0.9% NaCl to prepare 1 mg/mL solution on treatment days.

### 4.2. Cell Culture

The 4T1 triple-negative breast cancer (TNBC) cell line was provided by Judy Lieberman (Lieberman Laboratory, Harvard University, Boston, MA, USA). The 4T1 cells were cultured in Dulbecco’s Modified Essential Medium (DMEM) (4.5 g/L glucose, without L-glutamine and Phenol Red, cat no: DMEM-HXRXA, Capricorn Scientific, Ebsdorfergrund, Germany) supplemented with L-glutamine (200 mM) (Capricorn Scientific, Ebsdorfergrund, Germany, Cat-No. GLN-B), 10% Fetal Bovine Serum (FBS) (South America Origen, EU-approved, EuroClone S.p.A., Pero, Italy, Cat-No. ECS0180L), and a 10% Penicillin–Streptomycin mixture (Capricorn Scientific, Ebsdorfergrund, Germany, Cat-No. PS-B). The cells were kept at 37 °C in a humidified 5% CO_2_ incubator.

### 4.3. Animals

Six–eight-week-old female BALB/c mice were maintained under minimal disease (MD) conditions at the Animal Facility of the Basic Medical Science Center of Semmelweis University with free access to standard mouse chow and tap water ad libitum and under a 12 h dark/light cycle. Animals were housed and tested in full compliance with the Hungarian laws No. XXVIII/1998 and No. LXVII/2002 on the protection and welfare of animals, and with the directives of the European Union. All animal experiments were approved by the Pest County Government Office (PE/EA/50-2/2019).

### 4.4. mEHT Treatment

mEHT was performed using a LabEHY 200 device (Oncotherm kft., Budaors, Hungary) (Figure 8A) as described in detail earlier [20,28]. The LabEHY 200 generates an amplitude-modulated (AM), 13.56-radiofrequency electromagnetic field by the capacitive coupling between the position-adjustable upper electrode and the thermo-adjustable lower electrode. The lower electrode is connected to the LabEHY-200 with a radiofrequency (RF) cable and heating cable (Figure 8B). Thermal adjustment of the lower electrode serves to keep the animal’s body temperature in the physiologic range during narcosis. The LabEHY 200 device is connected to a computer; thus, the treatment parameters, such as the power, time, and modulation, can be adjusted by the LabEHY 200 controller software (v1.10, Oncotherm kft., Budaors, Hungary).

Mice anesthetized with 5% isoflurane were laid on the lower electrode, and the upper electrode was positioned on the tumor. mEHT was performed with 0.7 ± 0.3 watts for 30 min in a temperature-controlled manner after a 3 min long warm-up period. Four temperature sensors were used to monitor the temperature of the skin above the tumor, the rectum, the lower electrode, and the room (Figure 8C) [20,28]. The skin temperature was 40 °C during the treatment, resulting in a tumor temperature of 42.5 °C, as demonstrated previously [20,28]. The rectal temperature was 37 °C, and the temperature of the lower electrode was set to 37 °C during the 30 min treatment.

### 4.5. Inhibition of Tumor Growth

Orthotopic breast tumors were induced in female BALB/c mice by subcutaneous injection of 1 × 10^6^ 4T1 cells in 50 μL Phosphate-Buffered Saline (PBS) into the fat pad of the 4th mammary gland under isoflurane anesthesia (5% isoflurane for induction and 1.5–2% isoflurane to maintain anesthesia). On the eighth day after inoculation, the tumor size was measured using ultrasound (US) by visualizing the tumor in two perpendicular planes and measuring the length and width of the tumor (a,b). The (c) diameter was averaged from the depth of the tumor measured in the two positions. In addition, the digital caliper was used to measure three perpendicular diameters (a, b, c) of the tumor. The tumor volume (V) was calculated by the following formula: V = (a × b × c × π)/6, as described previously by Danics et al. [28].

We performed two experiments. The aim of the first experiment was to compare the effects of mEHT+LTLD to the other DOX formulations. Thus, 8 days after the implantation of 4T1 cells, the mice were randomly divided into 7 groups based on tumor volume and body weight: sham; mEHT; DOX; PLD; mEHT+DOX; mEHT+PLD; and mEHT+LTLD (*n* = 5–6). As LTLD is used in the clinic only in combination with hyperthermia [16,62,63,64], we omitted the LTLD+sham group from this study (Table 1).

Based on the results of our pilot study, we chose the 7.5 mg/kg dose of DOX from the three previously tested doses (5, 7.5, and 10 mg/kg) in the present study. In the pilot study, DOX reduced the body weight by more than 20% at the dose of 10 mg/kg, whereas 5 mg/kg did not inhibit tumor growth (Appendix A).

In the main study, DOX, PLD, and LTLD were injected at a dose of 7.5 mg/kg into the retro-orbital venous plexus, and the sham and mEHT groups received equivalent amounts of 0.9% saline. Based on our pilot study, LTLD was administered slowly over 3 min to avoid any hypersensitivity reactions. Immediately after the administration of the drugs, mEHT was performed for 30 min. The treatment was repeated 3 times at 72 h intervals (Figure 9). During the study, mice were monitored by measuring the tumor volume and body weight. Changes in body weight were used as an indicator of the systemic toxicity. Mice were sacrificed 24 h after the last treatment by cervical dislocation and the tumor tissues were resected. One half of the tumor was placed in 4% formaldehyde solution (Molar Chemicals Ltd., Halásztelek, Hungary) and transferred for histological processing. The other half was stored in liquid nitrogen at −80 °C for molecular analysis.

The purpose of the second experiment was to follow the growth of the tumors for 5 days after the third treatment to demonstrate when the tumors stopped shrinking. Based on the results of the first experiment, the mice were randomized into three groups: sham+vehicle (*n* = 5), mEHT+PLD (*n* = 7), and mEHT+LTLD. We followed the same protocol as in the first experiment.

### 4.6. In Vivo Optical Imaging of DOX Accumulation in Tumors

DOX accumulation in tumors was monitored using an in vivo imaging system (IVIS^®^ Lumina XRMS Series III Imaging System, PerkinElmer Inc., Waltham, MA, USA). Mice (*n* = 3) treated with 7.5 mg/kg DOX, PLD, or LTLD, with or without mEHT, were imaged at 1 and 24 h after injection. The DOX fluorescence signal was excited at 620 nm and captured at 500 nm with emission filters. Spectral unmixing was performed to subtract tissue autofluorescence. The acquired images were analyzed using Living Image^®^ 4.5 software (PerkinElmer, Inc, Waltham, MA, USA). The fluorescence intensity of the DOX was given in efficiency units defined as the ratio of the detected emission radiance (photons (p)/second (s)/steradian (sr)/area(cm^2^) over the excitation radiance (uW/cm^2^) [65]. The efficiency unit was quantified at the tumor site by marking a region of interest (ROI) that covered the tumor.

### 4.7. Histopathology and Immunohistochemistry

Tumor tissue samples were fixed in 10% neutral-buffered formalin prior to their embedding in paraffin (FFPE). FFPE samples were sectioned in 2.5 µm sections on a microtome at room temperature. The serial sections were mounted on glass slides and placed in a thermostat set to 65 °C for 1 h. Hematoxylin–eosin (H&E) staining was conducted by the deparaffinization and rehydration of the tumor sections in xylene and descending concentrations of ethanol, respectively. The rehydrated tumor sections were stained with hematoxylin (#05-M06002, Bio Optica, Milano, Italy) for 10 min, washed with tap water, and blued for 10 min. The eosin solution (#05-M10007, Bio Optica, Milano, Italy) was applied for 5 min. The stained sections were dehydrated with ascending concentrations of ethanol and xylene and mounted with a mounting medium (Bio Mount HM, #05-BMHM508, Bio Optica, Milano, Italy). H&E-stained sections were scanned and analyzed using Case Viewer image-analysis software (v.2.4, 3DHISTECH, Budapest, Hungary). The analysis was developed by a pathologist [66] with extensive experience in H&E staining, who trained one of our coauthors (Cs. S) to perform the evaluations. As demonstrated in Appendix A, the damaged area can be clearly delineated from the viable area. We annotated the damaged areas from the viable areas under high digital magnification in the case of each scanned tumor sample. The tumor destruction ratio (TDR) (%) was calculated by dividing the damaged area by the whole tumor area.

IHC for cleaved caspase-3 (cC3) and Ki67 was performed using a polymer–peroxidase system (Histols, Histopathology Ltd., Pécs, Hungary), as described previously [28]. Briefly, deparaffinized and rehydrated tumor sections were incubated for 15 min with 3% H_2_O_2_ in methanol for blocking endogenous peroxidases. Antigen retrieval was performed by heating the slides for 20 min in Tris–EDTA (TE) buffer (pH 9.0), using an Avair electric pressure cooker (ELLA 6 LUX (D6K2A, Bitalon Kft, Pécs, Hungary). For blocking nonspecific proteins, slides were incubated for 20 min in 3% bovine serum albumin (BSA) (#82-100-6, Millipore, Kankakee, IL, USA) diluted in 0.1 M Tris-buffered saline (TBS) (pH 7.4) and supplemented with 0.01% sodium azide. Primary antibodies were diluted in 1% BSA/TBS + TWEEN (TBST) (pH 7.4) (Table 2). Tumor sections were placed in a humidity chamber and incubated with the diluted primary antibodies overnight at room temperature. After washing, the sections were incubated with peroxidase-conjugated secondary anti-rabbit and anti-mouse IgGs (HISTOLS-MR-T, micropolymer-30011.500T, Histopathology Ltd., Pécs, Hungary) for 40 min. Washing steps after primary and secondary antibody incubations were performed using TBST buffer (3×, 3 min). A 3,3′-diaminobenzidine (DAB) chromogen/hydrogen peroxide kit (DAB Quanto-TA-060-QHDX-Thermo Fischer Scientific, Waltham, MA, USA) was used to detect the enzyme activity under microscopic control. After the dehydration and mounting steps, the slides were scanned, and the stained areas were evaluated digitally. Positive staining was evaluated using the QuantCenter module of Case Viewer by setting the intensity, color, and saturation in the annotated area. The cC3 expression was estimated in the whole tumor area and indicated as the relative masked area (the ratio of the masked area to the annotated area). In the case of Ki67 staining, the strong Ki67+ nuclei were counted in the annotated viable area. Due to staining problems, there are no data on Ki67 in three samples. Table 2 summarizes the antibodies and conditions used for the IHC.

### 4.8. Western Blot

Total protein was extracted from tumor tissues with TRIzol reagent (Molecular Research Center Inc, Cincinnati, OH, USA) according to the manufacturer’s instructions. Twenty micrograms of the protein extract were loaded per well, separated by 12% SDS-PAGE, and transferred onto a PVDF membrane. Tris-buffered saline (TBS), supplemented with 5% skim milk and 0.05% Tween 20, was used for blocking and antibody dilutions. The membrane was incubated with a primary antibody specific for cC3, Ki67, or β-actin, overnight at 4 °C (Table 2). After washing, the membrane was incubated with a horseradish peroxidase (HRP)-conjugated secondary antibody for one hour. The chemiluminescent signal was detected by an ECL Prime Western Blotting Detection Reagent (Cytiva, #RPN2232) and visualized with an Imager CHEMI Premium (VWR, Radnor, PA, USA). The blot was analyzed by Image J v1.53e software. The expressions of the proteins of interest were normalized to β-actin.

### 4.9. Statistical Analysis

Data are shown as means ± SEMs. GraphPad Prism software (v.6.01; GraphPad Software, Inc., La Jolla, CA, USA) was used for statistical analysis. Changes in the tumor volume, body weight, and DOX accumulation in the tumor were compared between groups by two-way ANOVA and Tukey’s post hoc test. The Kolmogorov–Smirnov test was used to test the normality of the data. The homogeneity of the variances among the groups was tested by Bartlett’s test. The data were normally distributed, and the variances across the groups were homogeneous. Therefore, one-way ANOVA, followed by Tukey’s post hoc test, were used to analyze changes in the tumor weights and expressions of cC3 and Ki67. The null hypothesis was rejected if * *p* < 0.05.

## 5. Conclusions

In conclusion, the present study revealed, for the first time, that mEHT can be used as a source of hyperthermia to activate thermosensitive nanoparticles, such as LTLD. In addition, mEHT improved the early tumor delivery and anticancer effects of DOX encapsulated in LTLD. However, the results of this study should be supported by further investigations on mEHT+LTLD in larger animals, such as pigs. As both mEHT and LTLD are under clinical trials, a clinical trial combining mEHT and LTLD in breast cancer patients is warranted.

## Figures and Tables

**Figure 1 ijms-25-03101-f001:**
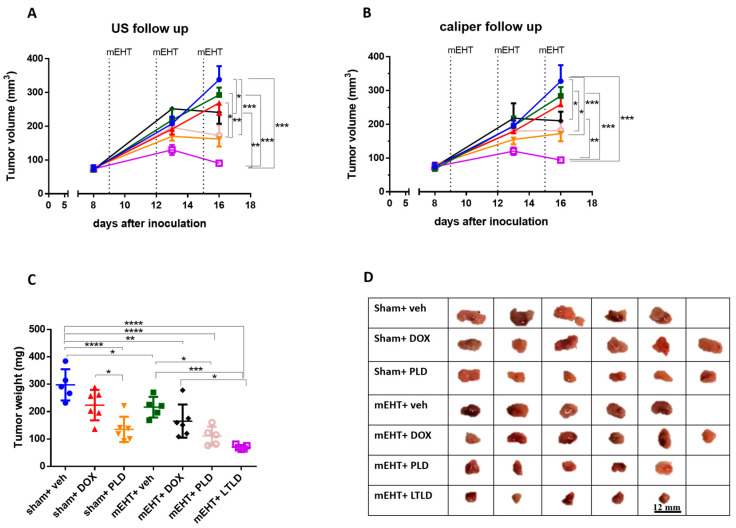
Effects of various treatments on tumor growth in 4T1-tumor-bearing mice. mEHT: modulated electro-hyperthermia; DOX: doxorubicin; PLD: PEGylated liposomal DOX; LTLD: lyso-thermosensitive liposomal DOX. (**A**) Ultrasound (US) and (**B**) digital caliper data after three mEHT treatments (marked with dotted lines). (**C**) Tumor weight. Data are means ± SEMs. (**D**) Scale images of all excised tumors. Each row shows all tumors in that group. (**A**,**B**) Two-way repeated measure ANOVA and Tukey’s post hoc test. (**C**) One-way ANOVA and Tukey’s post hoc test, *n* = 5–6/group, * *p* < 0.05, ** *p* < 0.01, *** *p* < 0.001, **** *p* < 0.0001.

**Figure 2 ijms-25-03101-f002:**
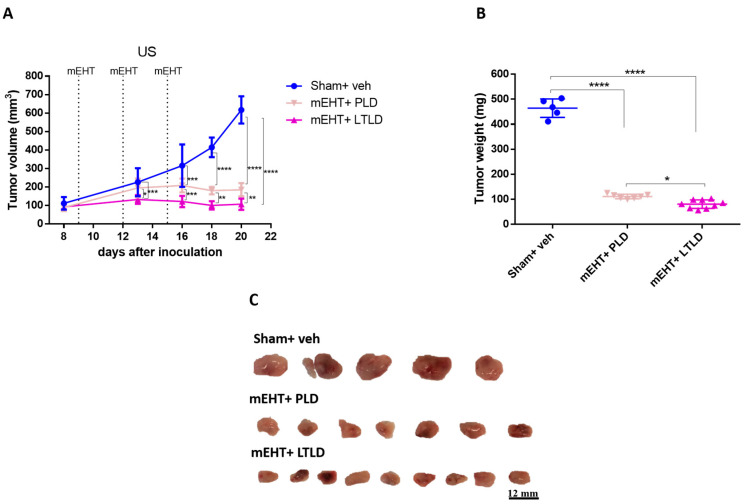
Effects of PLD and LTLD on tumor growth at 5-day follow-up after treatment termination in 4T1-tumor-bearing mice. (**A**) Ultrasound (US) results. (**B**) Tumor weight. (**C**) Scale images of all excised tumors. Each row shows all tumors in that group. Data are means ± SEMs. (**A**) Two-way repeated measure ANOVA and Tukey’s post hoc test. (**B**) One-way ANOVA and Tukey’s post hoc test, *n* = 5–9/group, * *p* < 0.05, ** *p* < 0.01, *** *p* < 0.001, **** *p* < 0.0001.

**Figure 3 ijms-25-03101-f003:**
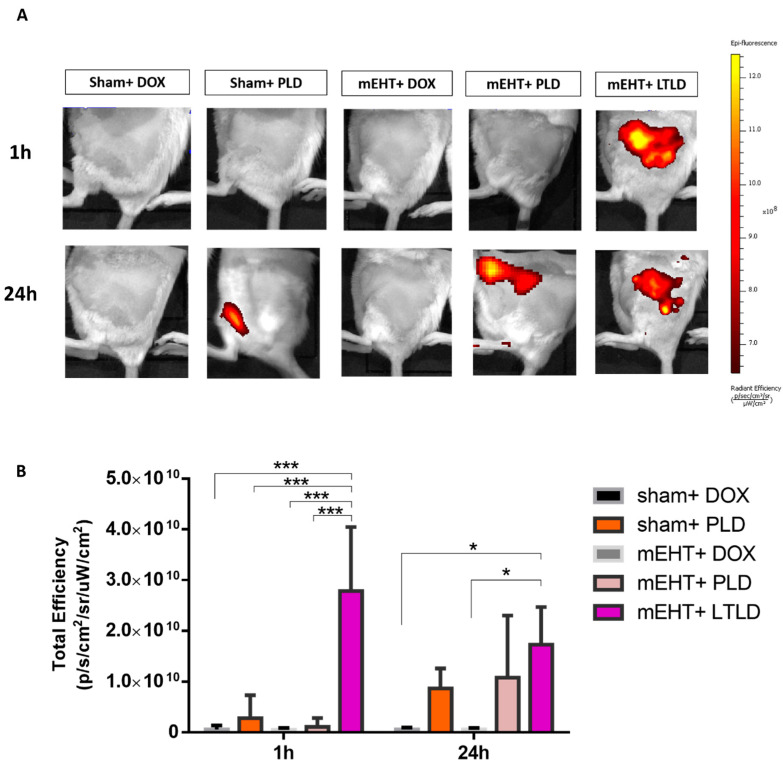
IVIS images of DOX accumulation in tumors at 1 and 24 h after treatment in 4T1-tumor-bearing mice. (**A**) Optical images of mice after treatment (yellow and red colors show DOX autofluorescence). (**B**) DOX fluorescence intensity quantified and expressed as total efficiency. Data are means ± SEMs. Two-way repeated measure ANOVA and Tukey’s post hoc test, *n* = 3/group, * *p* < 0.05, and *** *p* < 0.001.

**Figure 4 ijms-25-03101-f004:**
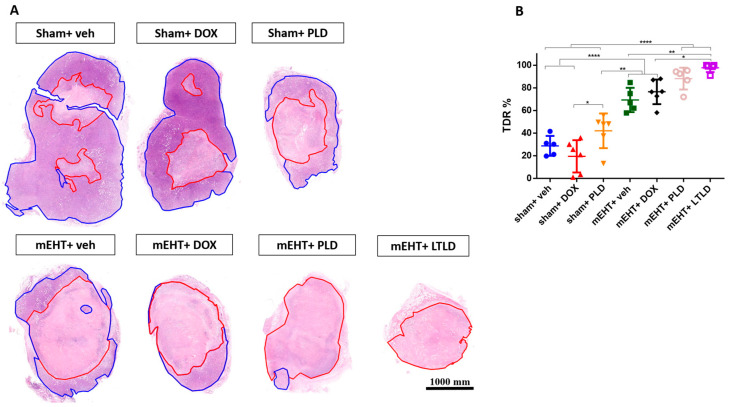
Tumor destruction ratio (TDR) 24 h after 3rd mEHT treatment. (**A**) Viable (blue) and damaged (red) areas are annotated in representative hematoxylin–eosin-stained sections. (**B**) TDR (%) estimated as percentage of damaged area to total surface area in hematoxylin–eosin-stained sections. Data are means ± SEMs. (**B**) One-way ANOVA and Tukey’s post hoc test, *n* = 5–6/group, * *p* < 0.05, ** *p* < 0.01 **** *p* < 0.0001.

**Figure 5 ijms-25-03101-f005:**
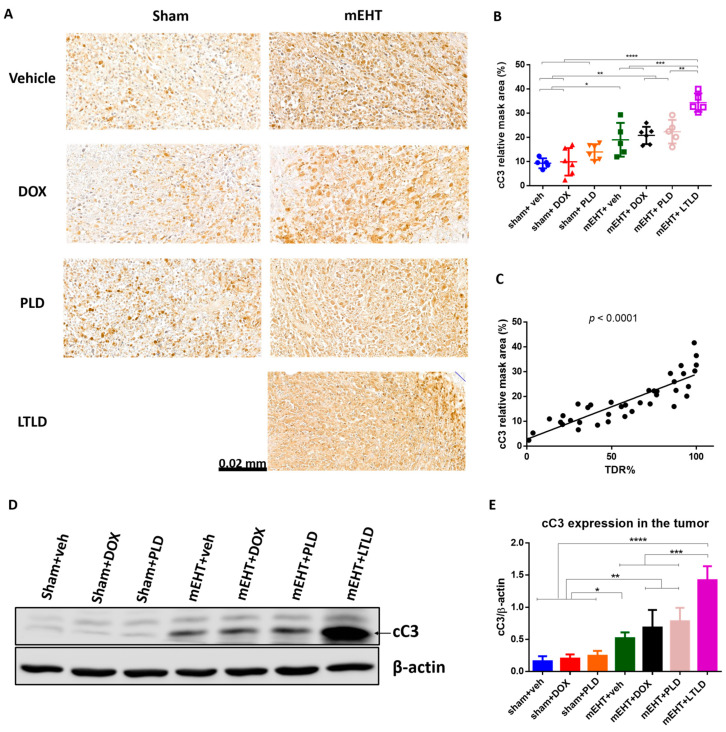
Cleaved caspase-3 (cC3) expression at 24 h after 3rd mEHT treatment. (**A**) Representative cC3-immunostained sections at 42× magnification. The cC3-positive tumor cells are stained brown. (**B**) Quantification of relative cC3 staining in tumors. (**C**) Correlation between cC3-stained area and TDR (%). (**D**) Representative images of Western blot. (**E**) Quantification of cC3 Western blotting. Data are means ± SEMs. (**B**,**E**) One-way ANOVA and Tukey’s post hoc test. (**C**) Linear regression, *n* = 5–6/group, * *p* < 0.05, ** *p* < 0.01, *** *p* < 0.001, **** *p* < 0.0001.

**Figure 6 ijms-25-03101-f006:**
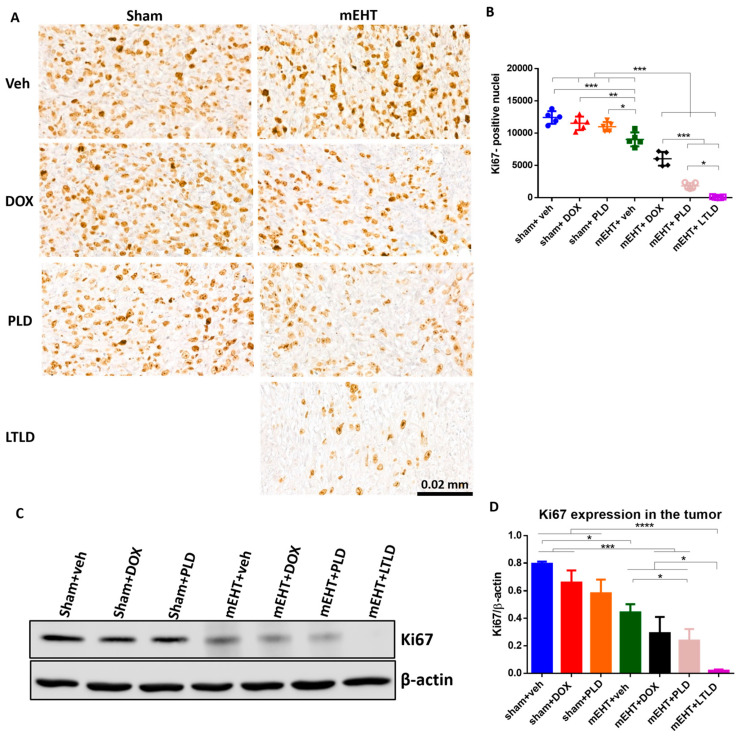
Ki67 expression in tumor at 24 h after 3rd mEHT treatment. (**A**) Representative images of Ki67-immunostained sections at 45× magnification. The Ki67+ nuclei are stained brown. (**B**) The number of Ki67+ nuclei. (**C**) Representative images of Western blot. (**D**) Quantification of Ki67 Western blotting. Data are means ± SEMs. (**B**,**D**) One-way ANOVA and Tukey’s post hoc test, *n* = 5/group, * *p* < 0.05, ** *p* < 0.01, *** *p* < 0.001, **** *p* < 0.0001.

**Figure 7 ijms-25-03101-f007:**
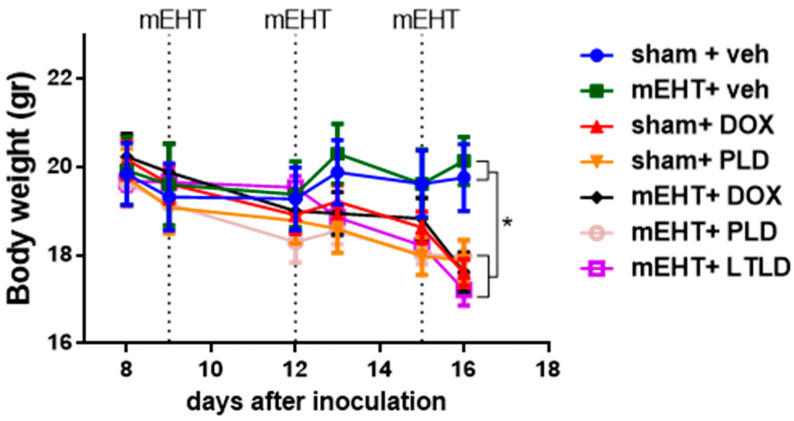
Effects of various treatments on body weights in 4T1-bearing mice. Data are means ± SEMs. Two-way ANOVA and Tuckey’s post hoc test, *n* = 5–6/group, * *p* < 0.05.

**Figure 8 ijms-25-03101-f008:**
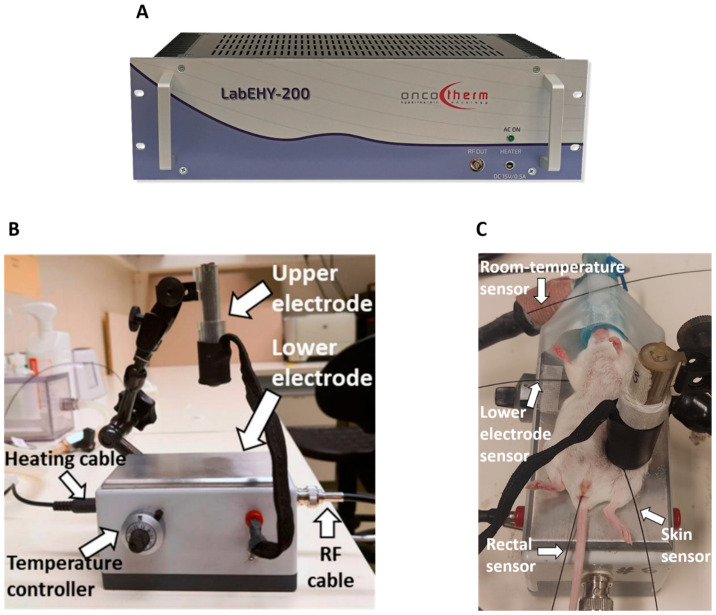
Modulated electro-hyperthermia (mEHT) mouse treatment device. (**A**) LabEHY 200 device. (**B**) mEHT treatment setup. (**C**) mEHT treatment of a mouse.

**Figure 9 ijms-25-03101-f009:**
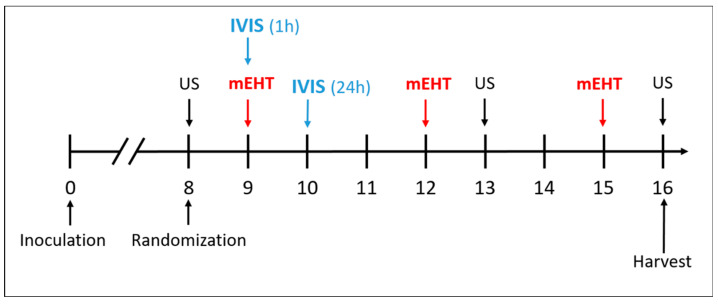
Schedule of tumor growth inhibition study.

**Table 1 ijms-25-03101-t001:** Study groups and number of mice in each group.

	mEHT	sham
Vehicle (saline 0.9%)	5	5
Free DOX	6	6
PLD	5	6
LTLD	5	-

**Table 2 ijms-25-03101-t002:** Antibodies and conditions used for immunohistochemistry (IHC) and Western blot (WB) ^1^.

Antigen	Type	Reference No.	RRID	Method	Dilution	Vendor ^2^
cC3	Rabbit, mAb	#9664	AB_2070042	IHC	1: 1600	Cell Signaling
				WB	1:1000	
Ki67	Rabbit, mAb	#MA5-14520	AB_10979488	IHC	1: 50	Invitrogen
				WB	1:100	
β-actin	Mouse, mAb	#ab6276	AB_2223210	WB	1:5000	Abcam

^1^ mAb: monoclonal antibody; cC3: cleaved caspase-3; Ki67: proliferation marker; RRID: Research Resource Identifier. ^2^ Vendor specifications: Cell Signaling (Danvers, MA, USA), Invitrogen (Waltham, MA, USA), Abcam (Cambridge, MA, USA).

## Data Availability

All data associated with this study are presented in the paper. The data that support the findings of this study are available from the corresponding author upon reasonable request.

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
