# Peer review of "Modulated Electro-Hyperthermia Accelerates Tumor Delivery and Improves Anticancer Activity of Doxorubicin Encapsulated in Lyso-Thermosensitive Liposomes in 4T1-Tumor-Bearing Mice"

_ijms, 2024, doi:10.3390/ijms25063101_

Round 1
Reviewer 1 Report
Comments and Suggestions for Authors
Dear authors,
I have recently reviewed your paper titled "Modulated Electro Hyperthermia Accelerates Tumor Delivery and Improves the Anticancer Activity of Doxorubicin Encapsulated in Lyso-thermosensitive Liposomes in 4T1-tumor bearing Mice" and I found it to be very interesting and well-written. Your study demonstrated that the use of LTLD+mEHT resulted in the highest accumulation of DOX in the tumor one hour after treatment, and also showed that tumor cell damage was associated with cleaved Caspase 3 in the damaged area and a reduction of Ki67 in the living area.
I have two questions regarding your study.
Firstly, I would like to know if the 4T1 cells reached other organs such as the liver or lymph nodes. Secondly, I suggest that the analysis of Caspase 3 and Ki67 should be performed by Western blotting to obtain exact quantification.
Author Response
Reviewer: I would like to know if the 4T1 cells reached other organs such as the liver or lymph nodes.
Author answer: Thank you for your question. In the Balb/C-4T1 model spontaneous metastasis do appear, however, not before 3 weeks has passed from the inoculation of 1 million (106) cells (Le MTN, Hamar P, et al: JCI. 2014/ fig 5: https://www.jci.org/articles/view/75695/figure/10). In the present, short-term study, no metastatic nodules were observed when the abdominal cavity of the mouse was opened. However, in one of our previous studies, we observed metastatic nodules in the lungs after one month of cell inoculation.
Reviewer: I suggest that the analysis of Caspase 3 and Ki67 should be performed by Western blotting to obtain exact quantification.
Author answer: Thank you for this suggestion. We performed a western blot for caspase3 and Ki67 and added the quantification to the result section. In addition, we added the western blot protocol to the methods section.
Reviewer 2 Report
Comments and Suggestions for Authors
In the article entitled “Modulated Electro Hyperthermia Accelerates Tumor Delivery and Improves the Anticancer Activity of Doxorubicin Encapsu lated in Lyso-thermosensitive Liposomes in 4T1- tumor bearing Mice” the authors provide a comprehensive and detailed analysis of the effects of Modulated electro-hyperthermia as a hyperthermia source for the release of Doxorubicin from LTLD in cancer.
The following are some suggestions that should be made to improve the manuscript.
Introduction
- Line 58 - insert parenthesis in the acronym “HEAT”.
Materials and Methods
- It is not sufficiently clear how the hematoxylin and eosin stain was done; the authors should provide more details.
- The authors should provide a more detailed description of how the immunohistochemical reaction was performed.
- Table 2. Authors should include the “Research Resource Identifiers” for each antibody used.
- The choice of the one-way ANOVA statistic must be supported by a previous analysis of the normality distribution of the data. Authors should provide additional detailed information on the choice of this type of parametric analysis.
Results
- Figure 3. Authors should include the scale bar in panel D
- Figure 4. Authors should include the scale bar in panel C
- Figure 6. Authors should include the scale bar in panel A
- The authors state that the main form of mEHT+LTLD-activated cell death is apoptosis. However. the use of a single marker (i.e. cC3) is not sufficient to exclude the involvement/activation of any other regulated cell death mechanism (e.g. necroptosis or autophagy). The authors should provide more results in order to validate their theory.
- Figure 7. The exact cellular localization of the investigated marker is not easily identified. The authors should include a higher magnification for the immunohistochemical reaction reported in panel A.
Author Response
The following are some suggestions that should be made to improve the manuscript.
Reviewer: Introduction
- Line 58 - insert parenthesis in the acronym “HEAT”.
Author answer: Thank you, we inserted parenthesis in the acronym “HEAT” in the revised manuscript (MS).
Reviewer: Materials and Methods
- It is not sufficiently clear how the hematoxylin and eosin stain was done; the authors should provide more details.
Author answer: Thank you for your suggestion. We included the detailed protocol for hematoxylin and eosin staining.
Reviewer: The authors should provide a more detailed description of how the immunohistochemical reaction was performed.
Author answer: Thank you we included more details about the immunohistochemistry staining.
Reviewer: Table 2. Authors should include the “Research Resource Identifiers” for each antibody used.
Author answer: We added the RRID for all antibodies in Table 2.
Reviewer: the choice of the one-way ANOVA statistic must be supported by a previous analysis of the normality distribution of the data. Authors should provide additional detailed information on the choice of this type of parametric analysis.
Author answer: Thank you for this suggestion. We tested the normality of the data using the Kolmogorov–Smirnov test. Moreover, Bartlett's test was done to check the homogeneity of variances among the groups. Thus, we modified the statistics section in the methods by adding these.
Reviewer: Figure 3. Authors should include the scale bar in panel D
Author answer: we modified figure. 3 and added the scale bar in panel D
Reviewer: Figure 4. Authors should include the scale bar in panel C
Author answer: we modified figure. 4 and added the scale bar in panel C
Reviewer: Figure 6. Authors should include the scale bar in panel A
Author answer: we modified figure. 6 and added the scale bar in panel A
Reviewer: The authors state that the main form of mEHT+LTLD-activated cell death is apoptosis. However. the use of a single marker (i.e. cC3) is not sufficient to exclude the involvement/activation of any other regulated cell death mechanism (e.g. necroptosis or autophagy). The authors should provide more results in order to validate their theory.
Author answer: In a previous study, we investigated the nature of cell death in more detail with several markers, in our model, however, we did not reach conclusive results. However, in our model cC3 is a reliable marker. The aim of the present study was not focused on the nature of cell death. Thus, we modified the conclusions drawn from the cC3 staining. In the result section the sentence: “The degree of cC3 (brown) staining was significantly correlated with TDR (R2= 0.73, p <0.0001) (Fig. 7C), suggesting that the main form of cell death was apoptosis” is now modified in the revised MS to: “The degree of cC3 (brown) staining was significantly correlated with TDR (R2= 0.73, p <0.0001) (Fig. 7C), suggesting that apoptosis was involved in cell death”.
In our previous publications (PMID: 32927720, PMID: 33917524), mEHT induced cC3 expression within the dead areas of the tumors. Thus, in the current study, we investigated whether apoptosis can be further induced when mEHT is combined with different formulations of doxorubicin. In the present study, we did not investigate other mechanisms for cell death, such as autophagy or necrosis. In hyperthermia treatment, necrosis is mainly activated at temperatures higher than 45 ̊C (PMID: 1977828 DOI: 10.1080/09553009014552221, PMID: 31159342 DOI: 10.3390/cancers11060764). Moreover, we did not study autophagy because in our previous preliminary studies (unpublished data), we did not observe a significant effect of mEHT on the gene expression of different molecules involved in autophagy such as Beclin1, Autophagy-related protein-3 (ATG-3), ATG-5, and sequestosome 1 (SQSTM1) (figures attached here). Moreover, our next-generation sequencing (NGS) (Raw RNA-Seq data are available at the European Nucleotide Archive: https://www.ebi.ac.uk/ena, under study accession number PRJEB43813) of mEHT-treated tumors did not show significant effects on autophagy molecules (PMID: 33917524). Even when we combined mEHT with chemotherapy such as Methotrexate (MTX), autophagy was not affected. We added this explanation to the discussion in the revised MS.
Figure. Effect of mEHT on the gene expression of molecules involved in autophagy in 4T1-isografts at 24 after the treatment. (A) Effect of mEHT on the gene expression of Beclin1, autophagy-related protein 3 (ATG3), and ATG5. (B) Effect of mEHT combined with methotrexate (MTX) on the gene expression of Beclin1 and ATG5.
Reviewer: Figure 7. The exact cellular localization of the investigated marker is not easily identified. The authors should include a higher magnification for the immunohistochemical reaction reported in panel A.
Author answer: we modified figure. 7, and added images with higher magnification

Round 2
Reviewer 1 Report
Comments and Suggestions for Authors
Dear Author,
I think that your work is ready to be published.
Well done!
Author Response
thank you!